# The Immunological Epigenetic Landscape of the Human Life Trajectory

**DOI:** 10.3390/biomedicines10112894

**Published:** 2022-11-11

**Authors:** Iva Juříčková, Michael Hudec, Felix Votava, Jan Vosáhlo, Saak Victor Ovsepian, Marie Černá, Valerie Bríd O’Leary

**Affiliations:** 1Department of Medical Genetics, Third Faculty of Medicine, Charles University, Vinohrady, 10000 Prague, Czech Republic; 2Department of Children and Adolescents, Third Faculty of Medicine, Charles University, Vinohrady, 10000 Prague, Czech Republic; 3Královské Vinohrady University Hospital, Vinohrady, 10034 Prague, Czech Republic; 4Faculty of Engineering and Science, University of Greenwich London, Chatham Maritime, Kent ME4 4TB, UK

**Keywords:** *HLA*, histone, methylation, *TINA*, *PARTICLE*, long non-coding RNA, ChIP

## Abstract

Adaptive immunity changes over an individual’s lifetime, maturing by adulthood and diminishing with old age. Epigenetic mechanisms involving DNA and histone methylation form the molecular basis of immunological memory during lymphocyte development. Monocytes alter their function to convey immune tolerance, yet the epigenetic influences at play remain to be fully understood in the context of lifespan. This study of a healthy genetically homogenous cohort of children, adults and seniors sought to decipher the epigenetic dynamics in B-lymphocytes and monocytes. Variable global cytosine methylation within retro-transposable *LINE-1* repeats was noted in monocytes compared to B-lymphocytes across age groups. The expression of the human leukocyte antigen (*HLA*)*-DQ alpha chain* gene *HLA-DQA1*01* revealed significantly reduced levels in monocytes in all ages relative to B-lymphocytes, as well as between lifespan groups. High melting point analysis and bisulfite sequencing of the *HLA-DQA1*01* promoter in monocytes highlighted variable cytosine methylation in children and seniors but greater stability at this locus in adults. Further epigenetic evaluation revealed higher histone lysine 27 trimethylation in monocytes from this adult group. Chromatin immunoprecipitation and RNA pulldown demonstrated association with a novel lncRNA *TINA* with structurally conserved similarities to the previously recognized epigenetic modifier *PARTICLE*. Seeking to interpret the epigenetic immunological landscape across three representative age groups, this study focused on *HLA-DQA1*01* to expose cytosine and histone methylation alterations and their association with the non-coding transcriptome. Such insights unveil previously unknown complex epigenetic layers, orchestrating the strength and weakening of adaptive immunity with the progression of life.

## 1. Introduction

While the newborn human relies on maternal immunity, young children cultivate an enhanced ability to resist infections. Ultimately, the healthy adult will be equipped with a diverse immune repertoire tailored to sense and defend against pathological signals to which it has been exposed to since birth [1]. Immune function will generally decline with age, eliciting a tapered response in the elderly compared to younger generations. This will be reflected in the reduced production rate and quality of the mature B and T lymphocyte pools in secondary lymphoid tissues from the onset of middle age [2].

The major histocompatibility complex (MHC), otherwise known as the human leukocyte antigen (HLA) system, represents two major classes (I and II) of heterodimeric surface receptors expressed on immune cells, including B-lymphocytes and monocytes [3,4]. Its primary role is to present antigens to T-lymphocytes resulting in cellular activation, clonal expansion, and the elicitation of an adaptive immune response [4,5,6]. *HLA* genes are the most polymorphic in the human genome and the first reported genes (*HLA-DQA1*, *HLA-DQB1*, *HLA-DRA1*, and *HLA-DDRB5*, etc.) to be associated with longevity in humans [7,8]. However, the influential role of *HLA* polymorphisms on extended lifespan has been a source of debate [9], given consideration for antagonistic pleiotropy, hybrid vigor and the presence or absence of deleterious recessive alleles within this genomic context [10,11]. This interpretation predicts that *HLA* alleles may offer positive and negative influences at different periods throughout life [10].

Epigenetic alterations occur during a lifespan and have been reported to impact immune function, especially in auto-intolerance [1]. However, global methylation of genomic long interspersed repetitive DNA cytosine levels has been noted to be inversely proportional to life duration in mice [12] and attributed to reduced DNA methyltransferase 1 activity with age [13]. Global histone 3 lysine 27 trimethylation levels associated with aging are contradictory [14,15], with calls to focus on locus and cell type-specific dynamics to understand better immune factors influencing life expectancy [16].

Our study addressed this issue by examining epigenetic methylation patterns in B-lymphocytes and monocytes from healthy children, adults, and seniors from a previously reported homogeneous control cohort from the Czech Republic [4,6]. While initially determining a global methylation differential in monocytes across these age groups, the focus was placed on cytosine methylation dynamics within the *HLA-DQA1*01* locus. Histone methylation enabled the identification of an associated novel long non-coding RNA *TINA* in adult monocytes, thereby introducing the potential for further layers of complexity on the epigenetic landscape in the normal lifespan.

## 2. Materials and Methods

### 2.1. Isolation of B-lymphocytes and Monocytes from Peripheral Blood

Written informed consent was obtained from all individuals involved in this study with approval from the Ethics Committee of the Institute for Clinical and Experimental Medicine, Charles University, Prague, CZ (approval number EK-VP/I5/0/2016). A total of 18 participants (*n* = 18) were enrolled in this study and categorized as normal healthy individuals with no known autoimmune disease, inflammation, or malignancy.

The participants were grouped according to age as children (*n* = 6; years of age = 6–10), adults (*n* = 6; years of age = 19–53), or seniors (*n* = 6; years of age 63–95). Peripheral whole blood was acquired from all participants by a phlebotomist based at the Department of Medical Genetics of the Third Faculty of Medicine, Charles University, Prague, CZ. B-lymphocytes (5 × 10^6^) or monocytes (7 × 10^6^) were extracted from whole blood (10 mL) by positive isolation using CD19^+^ (cat. # 11143D, Thermo Fisher Scientific, Waltham, MA, USA) or CD14 (cat. # 11149D, Thermo Fisher Scientific, Waltham, MA, USA) magnetic dynabeads (cat. # 11143D, Thermo Fisher Scientific, Waltham, MA, USA) respectively as per manufacturer’s instructions. Cells were counted using a Burker counting chamber (Paul Marienfeld GmbH and Co., Lauda-Konigshogen, Germany).

### 2.2. Genomic DNA Isolation from B-lymphocytes and Monocytes

This protocol followed the previously published simple salting-out method for genomic DNA extraction [17]. Genomic DNA concentration was obtained by spectrophotometric measurement of optical density at 260 nm with purity determination by 260/280 nm ratio measurement (i.e., ratio value = 1 indicative of high-quality genomic DNA with low salt content).

### 2.3. Quantification of Global DNA Methylation

The global 5-methylcytosine content of monocytes and B-lymphocytes was measured per manufacturer instructions using a global DNA methylation *LINE-1* kit (cat # 55017, Active Motif Inc., Carlsbad, CA, USA). Genomic DNA (1 μg) extracted from monocytes and B-lymphocytes (see above) was fragmented by incubation with M*se*I (37 °C for >6 h) in reaction buffer (1×) followed by heat inactivation (65 °C for 20 min) of enzymatic digestive activity. Mock digestion comparison reflected baseline digestion of 12.7 ± 2% decrease post M*se*I fragmentation. Methylated and non-methylated standards were prepared as directed in the kit instructions using provided reagents. DNA standards and M*se*I digested DNA (100 ng) samples were hybridized to a biotinylated human *LINE-1* consensus probe (0.5 pmol) corresponding to a human *LINE-1* transposon by denaturation at 98 °C for 10 min, incubation at 65 °C for 1 h followed by 25 °C for 20 min. The *LINE-1* probe has been designed to assay a 290 bp region of the *LINE-1* repeat element containing 88 cytosine residues, of which 12 are in a CpG context. Hybridized samples and standard controls were immobilized on a 96-well streptavidin-coated plate and incubated for 1 h at RT with mild agitation. After washing to remove unbound fragmented DNA, samples and standards were blocked for 30 min at RT before overnight exposure to mouse anti-5-methylcytosine at 4 °C with mild agitation. After rinsing, horseradish peroxidase-conjugated anti-mouse secondary antibody was applied for 1 h at RT. A colorimetric reaction was initiated and stopped according to manufacturer instructions. Absorbance was determined at 450 nm with 655 nm as a reference. The percentage ratio of 5-methylcytosine to detectable CpG residues in samples was extrapolated from the generated standard curve using the linear line equation.

### 2.4. Bisulfite Conversion of Genomic DNA

Unmethylated cytosine nucleic acid bases in genomic DNA (1 μg) underwent bisulfite conversion using an EpiTect bisulfite kit (cat. # 59104; Qiagen, Hilden, Germany) according to the manufacturer’s instructions. Briefly, genomic DNA was mixed with sodium bisulfate, and DNA Protect Buffer was formulated to prevent fragmentation. Cytosines were bisulfite converted by repeated thermal cycles of 60 °C and 95 °C incubations followed by de-sulfonation cleanup via buffer addition and spin-column extraction.

### 2.5. Total RNA Isolation from B-lymphocytes and Monocytes

Total RNA was isolated from B-lymphocytes and monocytes derived from children, adults, or seniors using a GenElute mammalian RNA isolation kit (cat. # RTN70, Sigma Aldrich, St. Louis, MO, USA). In brief, 2 × 10^6^ cells were disrupted in a lysis solution, and RNA was extracted with phenol/chloroform with ethanol precipitation. Solid phase filter cartridge RNA purification was conducted using appropriate washing solutions with final elution in ultrapure water (heated to 95 °C) with concentration and purity assessment using O.D. 260/280 ratio determination (NanoDrop 1000, Thermo Fisher Scientific, Waltham, MA, USA). Total RNA was stored at −80 °C. Total RNA (1 µg) was converted into first strand cDNA using standard protocol procedures (with the inclusion of random hexamers) and reagents (cat. # 18091050, Thermo Fisher Scientific, Waltham, MA, USA).

### 2.6. Real-Time PCR Quantification

The NCBI reference sequences for all reported genes and GenScript real-time PCR Taqman primer design software (http://www.genscript.com/cgibin/tools/primer_genscript.cgi (accessed on 4 April 2021) were utilized for assay design. Taqman gene expression assays with primers and probes were designed and purchased as previously reported for *HLA-DQA1*01*, *HLA-DQB6*01* [18], lncRNA *PARTICLE* [19], and lncRNA *TINA* (Appendix A). The endogenous housekeeping gene *PIPA* was used for normalization purposes. The reaction conditions for single gene assays were as reported [19]. StepOne™ Real-time PCR systems (Life Technologies, Carlsbad, CA, USA) enabled holding (50 °C, 2 min; 95 °C, 10 min) and cycling (95 °C, 15 s; 60 °C 1 min; 40 cycles). Cycle threshold values were extracted, and fold changes in gene expression were determined by 2 (^−ΔΔCt^). Samples from the various age groups were normalized to a value = 1 and relatively compared.

### 2.7. HLA-DQA1*01 Promoter Cloning and Sequencing

A 459-bp long sequence of the *HLA-DQA1*01* promoter was amplified by PCR (PlatinumTM Taq DNA Polymerase; Thermo Fisher Scientific, Waltham, MA, USA). The amplified target sequence was confirmed by 1% TAE agarose gel electrophoresis with purification using a MiniElute Gel Extraction kit (cat. # 28604; Qiagen, Hilden, Germany).

The purified *HLA-DQA1*01* promoter sequence was inserted into the pGEM-T easy vector (cat. # A1360; Promega, Madison, WI, USA) according to the manufacturer’s instructions. *Escherichia coli* JM109 strain cells were transformed for cloning. Colonies were confirmed by blue/white selection on agar plates containing ampicillin (cat. # A9518-5G; Sigma-Aldrich, St. Louis, MO, USA), X-Gal (cat. # B4252; 5-bromo-4-chloro-3-indolyl-β-d-galacto-pyranoside; Sigma-Aldrich, St. Louis, MO, USA), IPTG (cat. # I6758; isopropyl-β-d-thiogalactopyranoside; Sigma-Aldrich, St. Louis, MO, USA). Plasmids were isolated via a commercial kit (cat. # 12943; Qiagen, Hilden, Germany) according to the manufacturer´s instructions. The Sanger method was commercially performed for *HLA-DQA1*01* gene promoter sequencing (KRD, Prague, Czech Republic). The *HLA-DQA1*01* gene promoter from genomic DNA extracted from monocytes of children, adults, and seniors were analyzed and aligned in BioEdit software (version 7.2.5, Informer Technologies, Inc., Altamor Drive Los Angeles, CA, USA). The *HLA-DQA1*01:01:01:01* genomic reference sequence (Release 3.49.0) was obtained from the Immuno—Polymorphism Database—International Immunogenetics Project-Human Leukocyte Antigen (IPD-IMGT/HLA).

### 2.8. HLA-DQA1*01 Promoter MS-HRM Analysis

Bisulfite converted DNA of participants was analyzed by Methylation-Sensitive High-Resolution Melting (HRM) software version 3.0.1 on a StepOnePlus™ Real-Time PCR System (Thermo Fisher Scientific, Waltham, MA, USA). MeltDoctor™ HRM master mix (cat. # 4415440; Thermo Fisher Scientific, Waltham, MA, USA) was used for amplification and high-resolution melting according to the manufacturer´s instructions. Briefly, volumes and concentrations for one reaction were as follows: MeltDoctor™ HRM master mix 10 µL, forward primer (5 µM) 1.2 µL, reverse Primer (5 µM) 1.2 µL, genomic DNA (20 ng/µL) 1.0 µL, deionized water 6.6 µL—total reaction volume 20 µL (Appendix A). The CpGenome Human Non-Methylated DNA standard set (cat. # S8001U; Sigma-Aldrich, St. Louis, MO, USA) and CpGenome Human Methylated DNA standard set (cat. # S8001M; Sigma-Aldrich, St. Louis, MO, USA) were used as 0% methylated and 100% methylated standards respectively. Samples were analyzed by StepOne software version 2.3 (Thermo Fisher Scientific, Waltham, MA, USA). Difference plots were set by HRM software version 3.0.1 (Thermo Fisher Scientific, Waltham, MA, USA), and samples were compared according to their difference values versus methylated standards. Sample curves were subtracted from a 50% reference curve to accentuate differences between similar melt curves as recommended (Applied Biosystems, Waltham, MA, USA).

### 2.9. Histone 3 and Histone 3 Lysine 27 Trimethylation (me3) Immunofluorescence

This protocol follows what was previously reported [20]. Monocytes were fixed upon exposure to 4% paraformaldehyde for 1 h, washed for 5 min with 1 × PBS, and permeabilized in 1 × PBST (1 × PBS including 0.5% Triton™ X-100 (cat. # X100–5ML, Sigma Aldrich, St. Louis, MO, USA) for 30 min. Following one wash for 5 min in 1 × PBS, monocytes were transferred to a blocking solution (1 × PBS containing 2% goat serum, 5% bovine serum albumin, and 0.5% Triton™ X-100) for 1 h at RT. Monocytes were subsequently exposed to an antibody representing rabbit anti-tri-methyl-histone 3 (Lys27) (cat. # PA5-31817, Thermo Fisher Scientific, Waltham, USA, 1: 200 in blocking solution) with o/n incubation at 4 °C. Monocytes were washed three times for 15 min in 1 × PBS and incubated in Alexa fluor^®^ 568 goat anti-rabbit IgG (H + L) (1:500; in blocking solution) for 1 h at RT in the dark. Monocytes were then exposed to mouse monoclonal anti-histone H3 (cat # ab10799, Abcam, Cambridge, UK; 1 in 100 dilution) with o/n incubation at 4 °C. Following washes as indicated above, monocytes were incubated with Alexa fluor^®^ 488 goat anti-rabbit IgG (H + L) (1:500; in blocking solution) for 1 h at room temperature in the dark and allowed to air dry. To prepare for microscopy, cells were mounted in VECTASHIELD™ HardSet™ (cat. # H1500, Vector Laboratories, Newark, NJ, USA) and visualized using an epifluorescence microscope (Zeiss AxioVision, White Plains, New York, NY, USA).

### 2.10. Protein Extraction

Protein extraction from B-lymphocytes and monocytes was performed using a T-PER reagent (cat. # 78510, Thermo Fisher Scientific, Waltham, USA) with the addition of protease inhibitor cocktail tablets (cat. # 04693116001, Roche, Basel, Switzerland). Protein concentration was determined using a bicinchoninic acid (BCA) assay (cat. # 23227, Thermo Fisher Scientific, Waltham, MA, USA).

### 2.11. Electrophoresis and Western Blotting

Cell lysates (25 μg, 10 μL) were mixed with 4 × NuPage LDS Sample Buffer (Life Technologies, Carlsbad, CA, USA; 2.5 μL) and heated for 5 min at 70 °C before loading onto 12% Bis-Tris NuPage gels (cat. # NP0342BOX, Novex, Life Technologies, Carlsbad, California, USA) with electrophoresis in 1 × MOPS-SDS running buffer. Detection of H3 or H3K27me3 was determined as indicated above. Following extensive washing with TBS-T, Nytran membranes were exposed to alkaline phosphatase-conjugated goat-anti-mouse (cat. # A-3562, Sigma Aldrich, St. Louis, MO, USA; 1 in 5000 dilution) or goat anti-rabbit (cat. # A-3687, Sigma Aldrich, St. Louis, MO, USA; 1 in 5000 dilution) secondary antibody for 1 h at RT. Specific bands were visualized using a mix of 5-bromo-4-chloro-3′-indolyl-phosphate p-toluidine and nitro-blue tetrazolium chloride solution (cat. # 1001973039, Sigma Aldrich, St. Louis, MO, USA). Western blots were photographed using a FluorChem HD2 gel visualization system (Alpha Innotec, Kasendorf, Germany) with specific methylation modification or protein band intensities quantified using ImageJ (NIH, Bethesda, MD, USA).

### 2.12. Chromatin Immunoprecipitation (ChIP)-qPCR

Monocytes were fixed in formaldehyde at 37 °C for 30 min and lysed in SDS lysis buffer (cat. 20-163, Sigma Aldrich, St. Louis, MO, USA). Chromatin was sheared by sonication to fragment lengths of 200–1000 bps (Bioruptor, Diagenode, Seraing, Belgium). Chromatin immunoprecipitation followed the instructions and reagents of a ChIP assay kit (cat. # 17-295, Upstate Biotechnology, Charlottesville, VA, USA). Following brief centrifugation to remove cell debris, the chromatin supernatant was diluted in a buffer containing protease inhibitors. Following the addition of Protein A agarose and salmon sperm DNA, samples were incubated at 4 °C with agitation for 30 min. Brief centrifugation enabled pelleting of the chromatin in agarose. In brief, sheared chromatin was incubated with anti-H3K27me3 antibody (cat. # PAB24549, Abnova, Taipei, Taiwan) o/n at 4 °C. Protein A agarose beads were added with 1 h incubation at 4 °C to collect the antibody/chromatin complex. After elution with buffer, reverse crosslinking was performed at 65 °C for 4 h. Chromatin (histone and associated nucleotides) was purified by phenol/chloroform extraction and analyzed by Western blotting for H3K27me3 modification and quantitative real-time PCR for long non-coding RNA. The primers and probes for qPCR amplification are listed in Appendix A.

### 2.13. Secondary Structure Prediction and Conservation of Long Non-Coding RNA TINA

The *LOC124901301* nucleotide sequence (NC_000006.12) was uploaded in FASTA format onto the RNAfold web server (http://rna.tbi.univie.ac.at/(accessed on 24 May 2022)). The minimum free energy (MFE) prediction was calculated as previously reported [21], and a color-annotated secondary structure drawing was generated using equilibrium base pairing probability measurements. A local consensus RNalifold structure was generated from aligned *TINA*: NC_000006.12 (2000 bp total capability) and *PARTICLE*: NR_038942.1 and sequences using LOCARNA (http://rna.informatik.uni-freiburg.de (accessed on 30 May 2022)). Compatible base pairs are colored in accordance with the LOCARNA online color index [22,23,24].

### 2.14. Statistical Analysis

Data comparison was performed using unpaired Student’s *t*-test in GraphPad PRISM 3.0 or Microsoft Excel software. Variation was represented as standard error. Values of *p* ≤ 0.05 were considered significant.

## 3. Results

### 3.1. Monocytes Reveal Variation in Global Methylation of Genomic DNA across Age Groups as Determined from Long Interspersed Nucleotide Element 1 Repeats

Expression of *long interspersed nucleotide element 1* (*LINE-1*) is tightly repressed in most somatic tissues by various biological mechanisms, including methylation [25], to prevent DNA damage and ensure genomic integrity [26]. While ~500,000 copies of *LINE-1* exist throughout the human genome, most are incapable of retro-transposition due to mutations, resulting in less than 100 which are fully competent [27]. This study investigated the percentage of global content of 5-methylcytosine within CpG islands of retro-transposable *LINE-1* sequences within B-lymphocytes and monocytes extracted from children, adults, and seniors. The 5-methylcytosine/CpG island percentage was 30.2 ± 3% in B-lymphocytes, with no significant difference between age groups (Figure 1A). In contrast, when the 5-methylcytosine/CpG island percentage was likewise compared in *LINE-1* within genomic DNA from monocytes, highly significant differences were observed between groups (Figure 1B). Global levels of 5-methylcytosine within *LINE-1* per CpG island were 17.2 ± 5% in monocytes extracted from children. Such levels increased significantly by 2.2-fold in adults compared to children (55.4 ± 4%; *p* = 0.0003; Figure 1B). Variable global 5-methylcytosine became further evident with lower levels found within monocytes from individuals of senior age compared to adults (6.6 ± 3%; *p* = 6.5 × 10^−5^). Reduced 5-methylcytosine modification of CpG islands within *LINE-1* elements supports previous findings of hypomethylation with aging and reflects a loosening of expression inhibition in the monocytes of seniors [28]. Highly elevated levels of the 5-methylcytosine modification of CpG islands within *LINE-1* sequences may indicate hormonal influence not evident in pre-pubertal children [29] and tighter epigenetic restriction ensuring genomic integrity within adult monocytes.

### 3.2. HLA-DQA1*01 and HLA-DQB1*06 Elicit Distinctive Expression Profiles in Monocytes Relative to B-lymphocytes across Age Groups

We previously reported on the interallelic differences of *HLA-DQA1* and *HLA-DQB1* expression in monocytes and B-lymphocytes [30]. Intriguingly, it was noted that *HLA-DQA1*01* and *HLA-DQB1*06* showed low expression levels compared to other alleles in monocytes and B-lymphocytes. This study chose to focus on these specific alleles with the view that more stringent epigenetic methylation influences potentially elicited their reduced expression. Significantly lower levels of *HLA-DQA1*01* in monocytes were noted for all age groups (child: *p* = 0.015; adult: *p* = 2.6 × 10^−6^; senior: *p* = 1.8 × 10^−6^; Figure 2A) when relatively compared to B-lymphocytes. Adults showed the greatest reduction in *HLA-DQA1*01* expression in monocytes compared to children (*p* = 0.02) or seniors (*p* = 0.005) (Figure 2A). No significance was found in the level of *HLA-DQA1*01* in children versus seniors (*p* = 0.35) (Figure 2A). In addition, levels of *HLA-DQB1*06* were comparable between monocytes and B-lymphocytes as well as across age groups (Figure 2B). The variable *HLA-DQA1*01* transcription in monocytes extracted from children, adults, and seniors indicated potential flexibility in epigenetic regulatory components throughout the lifespan, prompting a further study of this immunological locus.

### 3.3. Dynamic HLA-DQA1*01 Promoter Methylation through Representative Lifespan Stages

Focus was placed on the promoter region of *HLA-DQA1*01* in monocytes to determine whether fluctuations occur in methylation levels between representative age groups. HRM analysis was undertaken post PCR and bisulfite sequence conversion of the *HLA*-*DQA1* promoter to discriminate genomic DNA sequences based on methylated cytosine content. Sample curves from children, adults, and senior monocyte genomic DNA were subtracted from the 50% methylation standard reference (Figure 3A–D). This enabled accentuated differences between melt curves to be visualized and analyzed in comparison to a reference standard range (Figure 3A). The percentage of methylation within the *HLA*-*DQA1*01* promoter was not significantly different when samples from children (45 ± 18%) and seniors (37 ± 12%) were compared with one-sided t-testing (*p* = 0.377) (Figure 3B,D). A significant reduction in the percentage of methylation within this genomic region was noted when adults were compared to children and senior cohorts (Figure 3B–D). HRM analysis revealed that methylation levels of the *HLA*-*DQA1*01* promoter were lower and less variable in adults (12 ± 0.4%) and compared significantly lower to children (*p* = 0.035) or seniors (*p* = 0.023). Herein, these findings highlight the dynamic nature of the epigenetic signature within the *HLA*-*DQA1*01* promoter, reflecting higher methylation levels at potentially vulnerable phases within the human lifespan (i.e., young and old stages) versus reduced levels during adulthood when optimum immune status has been generally acquired.

### 3.4. Identification by Bisulfite Sequencing of Methylated Cytosine Positioning within the HLA-DQA Promoter

We have previously reported a detailed methylation profile of the *HLA-DQA1*01* promoter in healthy adults [18]. Given the variable methylation level of the *HLA*- *DQA1*01* promoter in genomic DNA from monocytes extracted from children and seniors, the focus was placed on identifying the exact base positioning of this epigenetic modification. Targeted *HLA*-*DQA1*01* promoter amplification, bisulfite sequencing, and alignment identified the consensus W-, X-, and Y boxes [31] upstream from the transcription start site (Figure 3E,F). The methylation of CpG sites was identical except at −370, −307, and −192 positions which were non-methylated (Figure 3F). The significance of these epigenetic variation sites remains to be deciphered but may potentially alter the conformational landing platform for transcription factors.

### 3.5. Elevated Histone Modification in Adult Monocytes Associates with the Long Non-Coding Transcriptome

Epigenetic methylation of histone 3 lysine 27 (H3K27me3) is recognized as a modification marker of longevity [32]. Nevertheless, the cellular context and timing of H3K27me3 in the context of lifespan should also be considered [32]. This study examined H3K27me3 in monocytes extracted from children, adults, and seniors. Immunohistochemical analysis of monocytes enabled histone H3K27me3 modification to be visualized in adult samples (Figure 4A). However, no discernible signal was noted for H3K27me3 immuno-staining of monocytes from children or seniors. Further Western blot analysis did reveal evidence for H3K27me3 modification in monocytes in these age groups, albeit to a significantly lower level compared to adults. Results showed a 5.75-fold elevated H3K27me3 level in monocytes from adults compared to children (*p* = 0.002, Figure 4B). Likewise, a 3.09-fold increase in H3K27me3 was noted in adults compared to seniors (*p* = 0.006). Chromatin fractionation from adult monocytes and exposure to anti-H3K27me3 (Figure 4C,D), enabled the pulldown of the epigenetic modifier long non-coding RNA *PARTICLE* (Figure 4E). Interestingly, a signal was also detected for a previously uncharacterized long non-coding RNA *LOC124901301*, renamed in this study lncRNA *TINA* (*T*he *I*ntragenic long *N*on-coding RNA running *A*ntisense to *HLA-DQA1*). This recently annotated lncRNA is located on chromosome 6 (NC_000006.12; 32641310-32634991; 6320 bp in length), overlapping in an antisense direction to the *HLA-DQA1* promoter. No expression of *TINA* was detected in monocytes from children or seniors. RNAfold software enabled the secondary structure determination of *TINA* (Figure 4F). While LOCARNA analysis revealed considerable conserved structural alignment between *TINA* and *PARTICLE* (Figure 4F). The implications of such similarities in the context of immunological variation in humans are currently unknown and will form the basis for future research.

## 4. Discussion

Immunological protection develops in childhood, peaks in adulthood, and declines with advancing age. Herein, the dynamics of epigenetic modifications have been deciphered in immunologically relevant B-lymphocytes and monocytes across three representative age groups. While B-lymphocytes elicited stability in global methylation through retro-transposable *LINE-1* repeat profiling, significant alteration in this epigenetic readout was revealed in monocytes. Global methylation in monocytes showed significant progressive elevation from childhood to adulthood with a reduction upon reaching old age. Furthermore, methylation of the genomic *HLA* locus of immunity was examined, especially focusing on *HLA-DQA1*01* and *HLA-DQB1*06* given their previously reported low expression in comparison to other allelic variants [30] with presumable epigenetic influence over such repression. Similar *HLA-DQB1*06* expression levels were detected in B-lymphocytes and monocytes from all representative age groups, evoking stability and comparable regulatory control for this allelic variant. In contrast, *HLA-DQA1*01* elicited significantly different expression in monocytes across the lifespan, with the lowest values seen in adults compared to children and seniors. This was perhaps supported by lower variation in HRM readouts of the *HLA-DQA1*01* promoter and specific CpG island methylation seen in adult monocytes within this genomic region. Moreover, elevated H3K27me3 in monocytes from adults contrasted sharply with very low levels in children and seniors. The association of this epigenetic modification with the long non−coding transcriptome introduced *TINA* as a potential regulatory influence, given its recent annotation as an antisense transcript to the *HLA-DQA1* promoter. 

The monocyte pool of classical, intermediate, and non−classical subsets circulate in a dynamic equilibrium state during stable homeostasis [33]. Nevertheless, with aging, phenotypic alterations occur particularly in CD14+ CD16+ monocytes [34]. Recently, evidence has emerged that retro-transposable *LINE-1* activity influences and even promotes the process of aging and age-related diseases [35]. This appears to be linked to epigenetic effects associated with retro-transposition and immune activation due to retrotransposon nucleic acid detection, considered as ‘sterile’ inflammation [36]. The decrease in global CpG methylation at *LINE-1* repeats in monocytes from seniors compared to adults in this study is supported by others showing the tendency of cytosine methylation to decrease with aging, especially in gene-poor regions of the genome enriched in repetitive sequences [37].

This study focused on the epigenetic regulation of the promoter region of *HLA- DQA1*01* in monocytes to determine whether fluctuations occur in methylation levels between representative lifespan age groups, given the complexity of this genomic region and previous detailed investigations of *HLA-DR* [34]. Previously, hierarchical profiling of promoter DNA methylation in *HLA-DQA1* alleles has been reported along with a non-correlative relation with mRNA transcription [18]. HRM data reported in this current study offers support for that conclusion, given lower *HLA-DQA1*01* promoter cytosine methylation appearing in monocytes from adults and reduced transcript expression levels compared to children and seniors. As DNA methylation is an epigenetic mechanism capable of repressing gene expression, it could be speculated that the variation in methylated cytosines within the promoter region of representative age groups needs to be further strengthened by additional histone modification influences.

Heterochromatin formation can be instigated by epigenetic histone 3 lysine 27 trimethylation, evoking gene repression and immune cell differentiation in adults [38]. A unique histone modification landscape has been determined in neonatal monocytes, which is suggested to contribute to this period of heightened infection risk [39]. Despite increased levels of H3K27me3 with age in mammals [14,40], a decline in the fidelity of this modification has been demonstrated due to genomic—wide drifting of this repressive mark [41]. In this study, superior H3K27me3 levels in monocytes from adults were noted compared to children and seniors. Immunological detection of this modification was possible only with monocytes sourced from adults. This may reflect subtle ambiguities across assay methodology, considering weak Western band detection in samples from young and old individuals. Nevertheless, it may also be evidence of reduced or altered H3K27me3 fidelity in monocytes from children or seniors.

Long non−coding RNA (lncRNA) is a recently regarded epigenetic control mechanism for regulating gene expression. The lncRNA *PARTICLE* (Gene *PARTICL- ‘Promoter of MAT2A—Antisense RadiaTion Induced Circulating LncRNA*) serves to interlink epigenetic DNA and histone H3K27me3 modification mechanisms [20]. Despite this being the first report of *PARTICLE* expression in monocytes, it was expected that this lncRNA would be associated with H3K27me3 via chromatin immunoprecipitation, given previous reports [19,20]. This study amplified a recently annotated lncRNA *LOC124901301*, named lncRNA *TINA* in this study, after ChIP H3K27me3 pulldown. The limitations of this study include the fact that the full characterization of this novel lncRNA in the context of *HLA-DQA1* regulation is currently unknown. In addition, the study cohort was small and included a wide age group range of mixed gender.

In summary, it has been demonstrated that epigenetic control of the *HLA* gene locus in monocytes alters as individuals progress through life. The implications for health and disease susceptibility in children and seniors stress the importance of gene regulatory mechanisms beyond the heterogeneous allelic sequences of immunologically relevant loci.

## Figures and Tables

**Figure 1 biomedicines-10-02894-f001:**
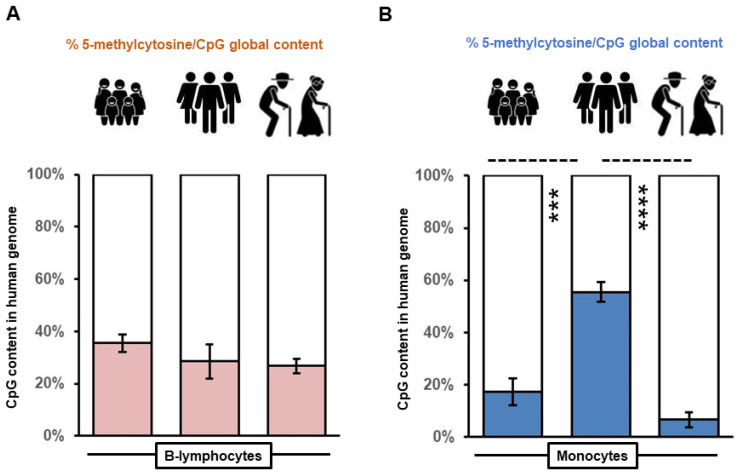
Monocytes reveal variation in global methylation of genomic DNA across age groups as determined from *long interspersed nucleotide element 1* (*LINE-1*) repeats. Histograms representing the percentage of 5-methylcytosine per CpG islands of retro-transposable *LINE-1* sequences within B-lymphocytes (**A**) and monocytes (**B**) extracted from children, adults, and senior individuals (*n* = 6 per group). Mean values are shown within the histogram rectangle with standard error bars indicative of variation between individuals. Asterisks represent significant comparative differences between groups indicated with dashed lines (*** *p* = 0.0003; **** *p* = 6.5 × 10^−5^).

**Figure 2 biomedicines-10-02894-f002:**
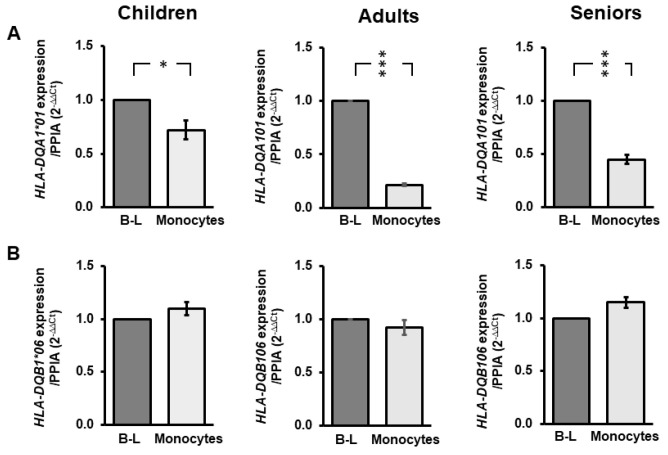
*HLA-DQ* allelic variants elicit alternative expression profiles in monocytes across age groups. Histograms of *HLA-DQA1*01* or *HLA-DQB1*06* expression in monocytes compared to B-lymphocytes (B-L) in children, adults, and seniors after normalization to the endogenous control gene *Peptidylprolyl isomerase A* (*PPIA*) showing differential (**A**) or stability (**B**) levels respectively between age groups. Asterisks represent significant within-group (*n* = 4) differences between B-lymphocytes and monocytes (* *p* = 0.01; *** *p* < 10^−6^).

**Figure 3 biomedicines-10-02894-f003:**
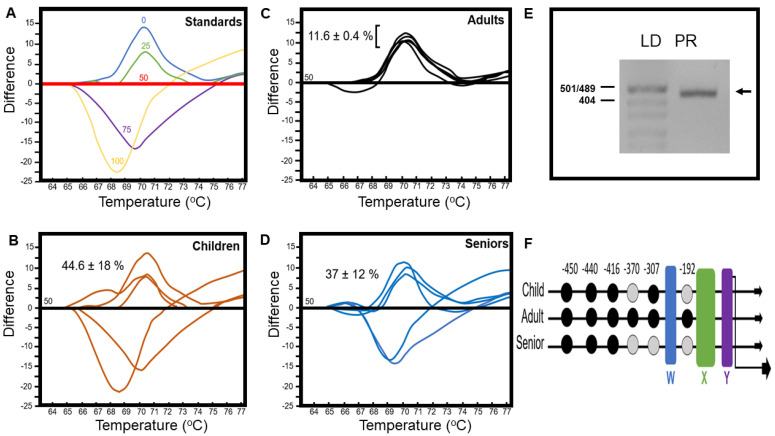
Dynamic *HLA*−*DQA1* class II promoter methylation throughout the lifespan. Representative standard high-resolution melting (HRM) curves relative to fifty percent methylation level (red line) (**A**). Numbers indicate percentage methylation values (**A**). Sample HRM curves of the *HLA-DQA1* promoter amplified from monocyte genomic DNA and subsequent bisulfite conversion (**B**–**D**). All melt curves were subtracted from the fifty percent reference standard to accentuate differences. Percentage methylation within the *HLA-DQA1* promoter of monocytes from children (**B**), adults (**C**), and seniors (**D**) are indicated along with standard error values. Values were determined from the melt curve peak, with above/below representing over/under 50 percent methylation values. Gel electrophoresis of PCR amplified *HLA-DQA1* promoter region. Lane 1: pUC19/M*sp*1 DNA marker (LD); Lane 2: *HLA-DQA1* promoter (PR; arrow, **E**). Schematic of cytosine positions within the *HLA-DQA1* promoter (numbers) showing differential methylation status between human lifespan age groups. Cytosines methylated (black circles), unmethylated (grey circles); regulatory domains W—box (blue), X—box (green), and Y—box (purple); downward arrow represents transcription start site (**F**).

**Figure 4 biomedicines-10-02894-f004:**
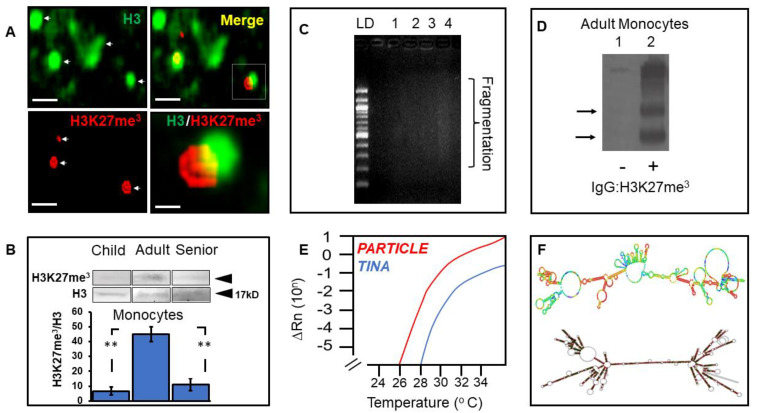
Monocytes from adults reveal elevated histone 3 K27 trimethylation and association with long non−coding RNAs *PARTICLE* and *TINA*. Epifluorescence micrographs of histone 3 (green) and lysine 27 methylation (red) in monocytes from adults. Scale bars 10 μm—upper and lower left panels; 2 μm—lower right panel (**A**). Merged images (yellow). Representative Western blot of H3K27me3 levels in histone 3 (H3) in monocytes extracted from a child, an adult, and a senior (above) with histograms showing group comparisons (below) (**B**). Asterisks represent significance between groups (*n* = 4; ** *p* ≤ 0.006). Gel electrophoresis of genomic DNA showing fractionation range after sonication. DNA 100 bp ladder (LD), fractionated genomic DNA after 5, 10, 15, and 30 s of sonication (Lanes 1–4 respectively) (**C**). Western blot after chromatin immunoprecipitation with/without anti−H3K27me3 (plus/minus). Arrows indicate the presence of histone 3 K27me3 in monocytes extracted from adults (**D**). Amplification curves of long non−coding RNAs *PARTICLE* and *TINA* following ChIP−RNA pulldown in monocytes from adults (**E**). Secondary structure of lncRNA *TINA* (left), comparative LOCARNA structural comparison with lncRNA *PARTICLE* (right)—dark bases indicative of stoichiometric similarity (**F**).

## Data Availability

The data presented in this study are available on request from the corresponding author.

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
