# Peer review of "The Immunological Epigenetic Landscape of the Human Life Trajectory"

_biomedicines, 2022, doi:10.3390/biomedicines10112894_

Round 1

Reviewer 1 Report

This is a very interesting article on epigenetics in B lymphocytes and monocytes in peripheral blood.

Some major aspects should be considered by the authors.

1. The sample size is low endangers the results obtained.

2. The authors should specify the age range that includes each group of the analysed cohort.

3. The entire document should be reviewed and type errors corrected.

4. Abbreviations in subsections should be avoided.

Reviewer 2 Report

“The Immunological Epigenetic Landscape of the Human Life Trajectory” by Juříčková et al is a somewhat interesting study interrogating epigenetic controls with DNA/histone methylation in two immune cell subsets in a cohort of 18 healthy individuals with an age range that spans from 6 to 95 years old. The manuscript is fairly well written. While the study appears to be reasonably well thought out with the use of somewhat appropriate methodologies, the main concern is its extremely small sample size (n = 18) for the outcomes to be interpretable, meaningful and free of sampling and statistical bias, and most importantly, for the representation of an age range that spans widely over 90 years. The study sample size must be significantly increased before a recommendation for publication can be made.

Furthermore, as the study outcomes and interpretation appear to be revolving around an HLA typing of HLA-DQA1*01 and HLA-DQB1*06, how was the HLA typing performed in these 18 individuals? What are their HLA typing across all relevant loci? Do they all have HLA-DQA1*01 and HLA-DQB1*06? The details of HLA are entirely missing in the manuscript.

Improper HLA nomenclature - There is no space in either side of the hyphen for the HLA gene names, i.e. HLA-DQA1, HLA-DQB1, etc, based on the proper nomenclature system established by the WHO HLA Nomenclature Committee. Also, there are several typographical errors – e.g. DDRB5, HLA – DQB5*01, HLADQ – A1, etc.

As it stands, over 4 pages have been allocated to the materials and methods section. While the reviewer appreciates the details and thoroughness, to improve the read and the flow of the manuscript and to keep the audience engaged, the reviewer strongly recommends that the materials and methods section be significantly shortened and made concise, at least for the main text. Some of the described methods, e.g. genomic DNA/RNA isolation, quantification, electrophoresis, Western blot, etc, are widely common processes. The details included do not necessarily add values or addition clarity to the manuscript. Please verify the statement “ratio value = 1 indicative of high - quality genomic DNA with low salt content”.

Limitations of the study are entirely missing in the discussion.

Reviewer 3 Report

This study reveals the epigenetic feature on the HLA-DQA1*01 locus in B cells and monocytes across different ages including children, adult and senior people. Although limited sample numbers were provided, this study still provides an interesting clue for the immunology-related epigenetic regulation. Here I have some suggestions:

1. This study focuses on the HLA-DQA1*01 locus in B cells and monocytes. The title seems too broad. A specific title related to this study should be provided. 

2. RNA and protein levels of HLA-II (HLA-DQ, DP, DR) of B cells and monocytes across age groups should be detected.

3. CD14+ monocytes were collected from PBMC and used for the analysis. However, there are different subpopulations in monocytes, including DC and macrophages. A flow characterization of these CD14+ monocyte subpopulations should be done to provide clearer information.

4. Figure 4A, the bar scale is missing.

5. There are some typos. Line 97-98 and Line 160-161, oC; Line 85, 5 × 106; Line 95, 20 1?

Round 2

Reviewer 1 Report

The article is correct in its present form.

Reviewer 3 Report

No more comment. The current manuscript could be accepted.